# Flavanol Polymerization Is a Superior Predictor of α-Glucosidase Inhibitory Activity Compared to Flavanol or Total Polyphenol Concentrations in Cocoas Prepared by Variations in Controlled Fermentation and Roasting of the Same Raw Cocoa Beans

**DOI:** 10.3390/antiox8120635

**Published:** 2019-12-11

**Authors:** Kathryn C. Racine, Brian D. Wiersema, Laura E. Griffin, Lauren A. Essenmacher, Andrew H. Lee, Helene Hopfer, Joshua D. Lambert, Amanda C. Stewart, Andrew P. Neilson

**Affiliations:** 1Department of Food Science and Technology, Virginia Polytechnic Institute and State University, Blacksburg, VA 24060, USA; kcracine@ncsu.edu (K.C.R.) wiersema@vt.edu (B.D.W.); elauren7@vt.edu (L.A.E.); andhlee@vt.edu (A.H.L.); amanda.stewart@vt.edu (A.C.S.); 2Plants for Human Health Institute, Department of Food, Bioprocessing and Nutrition Sciences, North Carolina State University, Kannapolis, NC 28081, USA or; 3Department of Food Science, Pennsylvania State University, University Park, PA 16801, USA; hxh83@psu.edu (H.H.); jdl134@psu.edu (J.D.L.)

**Keywords:** flavan-3-ol, procyanidin, α-glucosidase, melanoidin, Maillard reaction

## Abstract

Raw cocoa beans were processed to produce cocoa powders with different combinations of fermentation (unfermented, cool, or hot) and roasting (not roasted, cool, or hot). Cocoa powder extracts were characterized and assessed for α-glucosidase inhibitory activity in vitro. Cocoa processing (fermentation/roasting) contributed to significant losses of native flavanols. All of the treatments dose-dependently inhibited α-glucosidase activity, with cool fermented/cool roasted powder exhibiting the greatest potency (IC_50_: 68.09 µg/mL), when compared to acarbose (IC_50_: 133.22 µg/mL). A strong negative correlation was observed between flavanol mDP and IC_50_, suggesting flavanol polymerization as a marker of enhanced α-glucosidase inhibition in cocoa. Our data demonstrate that cocoa powders are potent inhibitors of α-glucosidase. Significant reductions in the total polyphenol and flavanol concentrations induced by processing do not necessarily dictate a reduced capacity for α-glucosidase inhibition, but rather these steps can enhance cocoa bioactivity. Non-traditional compositional markers may be better predictors of enzyme inhibitory activity than cocoa native flavanols.

## 1. Introduction

Cocoa beans (*Theobroma cacao*) are concentrated dietary sources of flavanols, a subclass of polyphenols, which are thought to be responsible for many of the bioactivities of cocoa [1,2,3,4,5]. (−)-epicatechin and (+)-catechin (which is epimerized to (−)-catechin during roasting) are the major monomeric flavanols in raw cocoa. Native flavanols in cocoa beans are approximately 58% procyanidins (PCs), or flavanol oligomers and polymers (monomeric residues linked via 4→β6 or 4→β8 bonds). These flavanols exhibit potent antioxidant and health-protective activities, including the modulation of oxidative stress and potentially reducing the risk of various chronic conditions, such as cardiovascular disease (CVD), type II diabetes mellitus (T2D), and different forms of cancer. Additionally, cocoa beans contain other bioactives, such as lipids, fiber, lignins, melanoidins (after roasting), methylxanthines, and other complex compounds that have not been extensively characterized. Beans contain approximately 55% fat, 16% fiber, 10% protein, and 3% ash, depending on variety. The health benefits associated with dietary cocoa are likely due to multiple bioactive compounds and their interactions, rather than one compound or class of compounds, because of the complex composition of cocoa and the reactions that occur during cocoa processing [6].

After harvesting, cocoa beans undergo a series of processing steps, including fermentation, drying, roasting, winnowing, and various other processes that may include pressing or alkalization, to produce a final product, such as cocoa powder or chocolate. These processes strongly influence the chemical composition of the product, with fermentation resulting in approximately 0–70% loss of total polyphenols and roasting costing an additional 15–40% loss [7,8,9]. Additionally, non-enzymatic browning reactions occur between native cocoa polyphenols and mono- or polysaccharides, proteins, and amino acids to produce Maillard reaction products, most notably melanoidins [10]. The widely-accepted paradigm is that preservation of native flavanols is critical for retaining bioactivity [7,9,11]. However, it is possible that reactions occurring throughout processing may generate processing-derived compounds with novel activities, such as lignin-like complexes and melanoidins, potentially preserving or even enhancing certain bioactivities as compared to the raw cocoa bean [7,12,13,14,15,16,17,18,19]. The levels and activities of these large, complex, and diverse compounds in cocoa are largely unknown due to analytical challenges (such as lack of authentic standards, structural complexity and size of potential products), low bioavailability of large complex compounds, and the continued research focus on small monomeric flavanols. 

In vivo, flavanols have highly variable systemic oral bioavailability, with absorption being inversely proportional to molecular weight and ~0% absorption for compounds ≥ tetramer [20,21,22,23,24]. Thus, the actions of these bioactives are likely to predominantly occur in the lumen and epithelium of the gastrointestinal (GI) tract, where the delivery of native flavanols and processing products is high and not subject to various barriers and metabolism/transport processes that reduce flavanol concentration and activity. Bioactivity exerted within the lumen and epithelium of the gut might play a key role in the mitigation and prevention of obesity and related conditions, such as T2D and CVD. Specifically, the inhibition of gut digestive enzymes to limit macronutrient digestion is a promising mode of bioactivity that does not require systemic bioavailability. Previous studies demonstrate that cocoa flavanols inhibit lipases, α-amylase, and α-glucosidase; α-glucosidase appears to be the most susceptible to cocoa inhibition [13,25]. α-glucosidase is a brush border enzyme that hydrolyzes starch and maltose into absorbable glucose [26]. The inhibition of α-glucosidase is a potential strategy for inhibiting or slowing blood glucose absorption in the context of glucose intolerance. Commercially available α-glucosidase inhibitors, such as acarbose, miglitol, and voglibose, come with high prices and various side effects (such as GI discomfort), therefore warranting the investigation of dietary flavonoids (from berries, red wine, green tea, cocoa, etc.) as potential inhibitors [16]. While isolated PCs are effective α-glucosidase inhibitors, we recently found that various cocoas were effective inhibitors of α-glucosidase, despite large reductions in native flavanols as a result of fermentation and roasting [13]. These results challenge the idea that losses of native flavanols inherently reduces all bioactivity of cocoa and, therefore, warrant further investigation into the factors that determine α-glucosidase inhibition. Specifically, the impact of cocoa processing on subsequent bioactivity and the identification of non-flavanol cocoa components with bioactivity are of interest. 

Fermentation and roasting are the logical steps in cocoa processing to manipulate in order to produce a final product with variable flavanol composition. Sourcing cocoas with the same bean origin and known processing history is impractical if not impossible, due to poor bean traceability, poor documentation, and varying practices of fermentation. A controlled model pilot-scale fermentation using a common starting material is necessary to conduct research regarding the impact of fermentation and further processing on bean composition and subsequent bioactivity [27,28,29]. Therefore, the main objectives of this work were to (1) evaluate the effect of extremes in fermentation and roasting on the composition of cocoa beans and powder, (2) determine how fermentation and roasting affect α-glucosidase inhibitory activity of cocoa powder, and (3) identify the compositional factors and processing conditions that optimize α-glucosidase inhibitory activity of cocoa. 

## 2. Materials and Methods

### 2.1. Chemicals and Standards 

Cargill, Inc. (Wayzata, MN, USA) generously provided raw unfermented cocoa beans, sourced from Hispaniola. The beans were stored in burlap sacks at 3.5 °C prior to use. LC-MS grade acetonitrile (ACN), LC-MS grade methanol (MeOH), citric acid, yeast extract, malt extract, calcium-lactate pentahydride, tween 80, sodium hydroxide, magnesium sulfate heptahydride, manganese sulfate monohydride, sucrose, glucose, fructose, calcium carbonate, agar, and water were obtained from Thermo-Fisher Scientific (Waltham, MA, USA). Glacial acetic acid, methanol, and acetone were obtained from VWR (Radnor, PA, USA). The standards of (−)-epicatechin (EC), (±)-catechin (C), and procyanidin B2 (PCB2) were obtained from ChromaDex (Irving, CA). Standards of procyanidin C1 (PCC1), cinnamtannin A2 (CinA2), and DP 5-9 purified from cocoa (purity: DP 3-5: 93–99%, DP6-9: 80–92%) were obtained from Planta Analytica (New Milford, CT, USA). Ammonium formate, Folin-Ciocalteu reagent, 4-dimethylaminocinnamaldehyde (DMAC), and α-glucosidase (from *Saccharomyces cerevisiae*) were obtained from Sigma-Aldrich (G5003, St. Louis, MO, USA). The solvents were ACS grade or higher.

### 2.2. Fermentation Model System and Processing

A partial factorial approach (Figure 1A) was employed to generate cocoa powders from the same batch of raw beans: fermentation (unfermented (UF), cool fermented (CF), and hot fermented (HF)) and roasting (unroasted (UR), cool temperature roasted (CR), and hot temperature roasted (HR)). Certain possible combinations were not evaluated due to cost constraints. 

#### 2.2.1. Fermentation 

The raw unfermented cocoa beans were rehydrated, fermented, and dried based on previously established pilot-scale cocoa fermentation protocols described by Racine et al. [30] and Lee et al. [31] with modifications. The unfermented cocoa beans (32 kg) were rehydrated in plastic fermentation boxes (polypropylene, Sterilite, Townsend, MA, USA) by submersion in distilled, deionized (DI) water (45 L) for 24 h at room temperature. The final moisture content of the beans after rehydration was between 35–50% (IR-120 Moisture Analyzer, Denver Instrument, Bohemia, NY, USA). After draining off the excess water, rehydrated beans (60 kg) were mixed with 60 L of simulated pulp media (Table 1) (three replicate fermentation boxes per fermentation run; approximately 20 kg rehydrated beans; and, 20 L simulated pulp media per box). Boxes were covered and placed inside a pre-heated (25 °C) incubator (Forma 29 cu ft Reach-In-Incubator, Model No. 3950, Thermo Fisher Scientific, Waltham, MA, USA). 

In total, four fermentation runs were conducted: two cool and two hot fermentation runs. Each fermentation run continued for a total of 168 h. For each run, the material was fermented in three separate boxes within the same incubator, with 20 kg rehydrated beans and 20 L simulated pulp in each box. The incubator set point was raised 3.5 °C/24 h for the cool fermentation runs and 6 °C/24 h for the hot fermentation runs, to final temperatures of 46°C (cool) and 60°C (hot) (Figure 1A). For all of the boxes in all fermentation runs, the beans were manually agitated for 5 min. every 8 h to ensure that the simulated pulp media was well-mixed and properly aerated. The pulp dissolved oxygen (DO) and pH values were monitored while using benchtop meters (Orion DO Probe 083005MD; Orion Versa Star Pro pH meter; Thermo Fisher Scientific, Waltham, MA, USA). Appendix A shows images of a representative fermentation batch over time. After 168 h of fermentation, the beans were drained to remove the remaining simulated pulp media, evenly spread onto baking sheets, and then oven dried (Rational, Germany; Blodgett, Burlington, VT, USA) at 65.5 °C for 24–26 h or until the moisture content fell below 8%. After drying, all of the beans from both cool runs were thoroughly commingled and stored at 3.5 °C until roasting, and the same was done for all beans from both hot runs. The beans that were subjected to the unfermented treatment were immediately oven dried following rehydration and draining, according to the procedures above. 

#### 2.2.2. Roasting and Further Processing 

Roasting was performed in a gas-fired drum roaster (180 kg capacity, Probat, Inc., Vernon Hills, IL) at a drum speed of 15 rpm in collaboration with Epiphany Craft Malt (Durham, NC, USA). Each unique fermentation/roasting treatment was separately roasted in batches of approximately 30 kg. The cool roasted treatment temperature was 120 °C and the hot roasted treatment temperature was 170 °C (Figure 1A). Roasting treatments were conducted for 20 min. each. After roasting, the beans were placed on a rotating cooling table and then stored at 3.5 °C until further processing. The beans were further processed into cocoa powder in collaboration with Blommer Chocolate Company (East Greenville, PA, USA). The beans were first winnowed to remove shells and ground into liquor. The liquor was then heated to approximately 200 °C and pressed (Cacao Cucino, Model No. 306487, Clearwater, FL, USA) for 133 min. to produce a solid cake that was then ground into a homogenous cocoa powder. All seven treatments were uniformly processed into cocoa powder. The powders were stored at −20 °C until further analysis. The moisture and fat content for liquors and cake (ORACLE Rapid Fat Analyzer, CEM, Matthews, NC, USA) and particle size of liquors (Microtrac S3500, Microtrac, Montgomeryville, PA, USA) was measured for each treatment per Blommer standard operating procedures. 

### 2.3. Fermentation Assays

The bean pH was determined every 24 h based on the method that was described by Racine et al. [30]. A representative cut test was performed on a sample of whole beans from each of the four fermentation runs (2 cool batches, 2 hot batches). Beans (*n* = 6) from each 24 h sampling period (0–168 h) were cut through the middle lengthwise so that color and quality could be assessed. This test is a standard assessment of post-fermentation bean quality and suitability to move forward in processing [32,33]. The cocoa bean fermentation index (FI) was measured every 24 h during fermentation based on the method of Romero-Cortes et al. [34] with minor modifications. Randomly selected whole cocoa beans (*n* = 3–5) were frozen with liquid nitrogen and ground into a powder in an electric spice grinder. The powder (50 mg) was mixed with 5 mL MeOH/HCl (97:3 *v*/*v*) and extracted at 4 °C for 16–18 h on a rotating platform. The samples were then centrifuged (5 min., 3500× *g)*, supernatant collected (300 µL), and absorbance measured at 460 nm and 530 nm using a 96-well microplate. The FI was calculated while using the equation below and all analyses were performed in analytical triplicate.
(1)$$FI=\frac{A_{460}}{A_{530}}$$

### 2.4. Polyphenol Extraction and Characterization 

Polyphenol-rich extracts were made from raw beans, the fermented cocoa beans, and powders for all seven treatments, as described previously [13,30]. The Folin–Ciocalteu colorimetric assay was used to approximate the total phenolic content of the freeze-dried cocoa extracts, and total flavanols were measured by the 4-dimethylaminocinnamaldehyde (DMAC) colorimetric assay, as previously described in Dorenkott et al. [12]. These values were expressed in mg Gallic Acid Equivalents (mg GAE)/g bean and mg PCB2/g bean, respectively. The mean degree of polymerization (mDP) of the flavanols was determined by thiolysis based on the protocol of Dorenkott et al. [12], with minor modifications. Cocoa monomeric flavanols and PCs (DP 1-10) were quantified by HILIC UPLC-MS/MS based on the method of Racine et al. [35]. Refer to Appendix A, including Appendix A, for full methodological details.

### 2.5. Melanoidin Dialysis 

The dialysis method that was proposed by Sacchetti et al. [36] was followed with modifications to isolate high and low molecular weight (HMW and LMW, respectively) extract fractions. Polyphenol-rich extracts were re-dissolved in extraction solution (70:28:2 acetone, water, acetic acid) at 40 mg/mL. Dialysis was performed in triplicate while using acidified methanol: water (60:40, 0.1% formic acid) as the dialysis solvent. For each replicate, 2.5 mL of dissolved extract (i.e. 100 mg) was placed inside presoaked dialysis tubing (8–10 kDa MW cutoff, Spectrum Spectra/Por Biotech-Grade RC Dialysis, Fisher) and clipped closed. This MW cutoff was chosen based on preliminary data (Appendix A) illustrating that the majority of early, intermediate, and final Maillard reaction products (MRP) are eluted from the 3.5–5 kDa and 8–10 kDa tubing, with very little in 20 kDa, followed by a significant increase in 50 kDa (see Appendix A). The tubing was submerged in 250 mL dialysis solvent and the beaker was stirred at 4 °C for 24 h and continuously sparged with N_2_. Following dialysis, the solvent outside the tubing and the components remaining within the tubing were separately rotary evaporated at 55 °C, frozen at −80°C, and then freeze-dried. Following freeze-drying, the solids were weighed, and yield was calculated. 

Fractions and undialyzed extracts were resolubilized in 0.05 M H_2_SO_4_ to a concentration of 0.15625 mg/mL (for detection of early MRPs in all samples), 2.5 mg/mL (for detection of intermediate MRPs in all samples), 5 mg/mL (for detection of late MRPs in LMW, <8–10 kDa, fractions), 2.5 mg/mL (for detection of late MRPs in HMW, >8–10 kDa, fractions), and 10 mg/mL (for detection of late MRPs in unfractionated extracts) to selectively quantify melanoidins/MRPs. Each diluted dialysate, non-dialyzable extract, and unfractionated extract was transferred (300 µL) into a UV-Star 96-well plate. The absorbance was read at 280 nm (early MRP), 360 nm (intermediate MRP), and 420 nm (late MRP). The MRP values are reported as absolute absorbance values (single dilution used for each fraction to facilitate direct comparisons of absorbance values: 5 mg/mL (<8–10 kDa), 2.5 mg/mL (>8–10 kDa), and 10 mg/mL (unfractionated extract)) due to the lack of a good analytical standard. 

### 2.6. In Vitro α-Glucosidase Inhibition Assay 

The powder extracts were screened for α-glucosidase inhibitory activity *in vitro,* as described previously [13]. A 0.1 M phosphate buffer (pH 6.9) was prepared in water with sodium phosphate monobasic anhydrous (8.05 g/L), and sodium phosphate dibasic anhydrous (4.67 g/L). pH was adjusted to 6.9 with 1 N NaOH. The extracts were diluted to 0.3125–8000 µg/mL in 10% DMSO. A negative control (no inhibitor, i.e. 0 µg/mL extract) was prepared with only 10% DMSO. The α-glucosidase solution (1 U/mL) and *p*-nitrophenyl α-D-glucopyranoside (*p*NPG, 1 mM) substrate solution were prepared in 0.1 M phosphate buffer. In 96-well plates (*n* = 6 wells/treatment), 50 µL of each working sample solution or negative control was mixed with 100 µL of α-glucosidase solution. The plate was then incubated at room temperature for 10 min., followed by the addition of 50 µL of *p*NPG solution to each well. Thus, the final concentration range of each inhibitor solution was 0–2000 µg/mL. The plate was then read at 405 nm. The negative control was 0 µg/mL extract and the positive control was acarbose in 10% DMSO (0–2000 µg/mL final concentration). The cocoa extracts were compared to the controls at each concentration and the values were expressed as % α-glucosidase activity while using the following equation: (2)$$\%\;\alpha -Glucosidase\;Activity=\left(\frac{\Delta A_{\mathrm{sample}}}{\Delta {Ā}_{\mathrm{negative}\;\mathrm{control}}}\right)\times \;100$$
where:
ΔA_sample_= the change in the individual absorbance value of the product of the inhibitor, substrate, and enzyme at each inhibitor dose before and after incubation; and,ΔĀ_negative control_= the average change in absorbance of the negative control (0 µg/mL) before and after incubation.

α-glucosidase enzyme inhibition by all extracts and the positive control were assessed at physiologically relevant doses. The highest concentration (2000 μg extract/mL) is equivalent to approximately 13,333 μg powder/mL in the intestinal lumen (i.e. 13,333 ppm), when accounting for 15% extraction yield from powder. At an estimated 2 L upper digestive volume, this would equal approximately 26.67 g of cocoa product, or just less than 1 square of baking chocolate (1 square ≈ 28 g). At a lower concentration such as 100 µg/mL (666.67 µg original product/mL in the intestinal lumen), this would be equivalent to 1.33 g of original cocoa product or approximately 0.0475 squares of baking chocolate, which is a very achievable and physiologically relevant amount. Alternatively, acarbose is typically administered at 50–200 mg per dose [37]. At a 2 L upper digestive volume these would translate to 25–100 µg/mL, a range assessed in this assay and further confirming the physiological relevance of our concentrations.

### 2.7. Data Analysis and Statistics 

All of the fermentation data (pH, FI, DO) within each treatment were analyzed by one-way ANOVA and when overall significance was detected, Tukey’s HSD post hoc test was performed to determine the differences between time point means. Compositional data for beans and powders were analyzed for significance of main effect and interactions by two-way ANOVA using type III sums of squares to account for the unbalanced data (due to the partial factorial design) and if overall significance between treatments was detected, post-hoc comparisons of least-squares means (lsmeans) was performed. The normality was visually checked for each variable and with the Shapiro–Wilk test, and, if needed, transformed via Log or Box–Cox prior to two-way ANOVA and post-hoc test. Median inhibitory concentration (IC_50_) values for α-glucosidase were calculated for each extract treatment while using a four-parameter sigmodal analysis. Enzyme inhibitory data were analyzed using a four-parameter log-logistic model for each of the seven treatments and the positive control, acarbose. Simple linear regression analysis was performed to correlate individual measured compositional factors (predictors: X) to α-glucosidase activity (IC_50_, outcomes: Y). The mean compositional values and IC_50_ for each extract treatment were plotted (seven points per analysis), and *R* and *R*^2^ were calculated. Statistical significance for all of the treatments in this study was defined *a priori* as *p* < 0.05. All of the analyses were carried out on GraphPad Prism v7.03 (GraphPad, La Jolla, CA, USA), R (v3.5.2) with Rstudio (v1.1.463, Boston, MA, USA) and additional packages, including lsmeans, car, RVAideMemoire, and others, to assist in the dose-response analyses. 

## 3. Results

### 3.1. Fermentation Model System and Cocoa Processing

Monitored fermentation parameters (pH, DO) similarly progressed across 168 h for CF and HF, with initial (0 h) pulp pH values at 3.99 ± 0.05 and 3.84 ± 0.01, respectively, and ending (168 h) values at 3.51 ± 0.02 and 3.36 ± 0.07, respectively (Figure 1D). Furthermore, the bean pH values ranged from an initial 5.61 ± 0.01 (CF) to 5.75 ± 0.16 (HF) and concluded at 3.89 ± 0.04 (CF) and 3.95 ± 0.01 (HF) after 168 h. DO remained ≤1 mg/L after the first 24 h of fermentation. The initial FI values (Figure 1B) for both CF and HF were above 1.0 and remained within ±0.01 throughout the entire 168 h fermentation. Cut test results showed similar results, as there was no true progression of color from purple to brown beans. However, the sample size used was much smaller than what is traditionally seen, with the typical cut tests consisting of over 300 cut beans.

Appendix A reports the fat and moisture content of liquors and presscakes, as well as the particle size of liquors. While liquor fat content was generally similar across all treatments (~56–58%), UF/HR had much higher cake fat content (15.4%) and lower cake moisture (2.0%) than the other treatments. HF/HR had the lowest cake fat content, at 7.50%. 

### 3.2. Characterization and Quantification of Polyphenols

Folin–Ciocalteu and DMAC, respectively, measured the total polyphenols and flavanols in beans (Appendix A) and powders (Figure 1E) from each treatment. The powders had a 2–3-fold higher total polyphenol and flavanol enrichment as compared to beans due to the concentration of polyphenols during pressing as cocoa butter is removed. Overall, treatment significantly influenced the measured total polyphenols and total flavanols (Figure 1E). Treatments with less harsh (i.e. cooler) processing conditions (UF/UR, UF/CR, CF/CR) generally had higher levels of total compounds than those that endured a more heat intensive processing (HF/UR, HF/HR, UF/HR), as expected, with the notable exception of CF/UR, which did not fit the overall pattern. 

PCs (DP 1–10) were quantified from beans and powders from each of the seven treatments via HILIC UPLC-MS/MS. The powder concentrations were normalized to the fat-free mass in each treatment to account for the effect of pressing (i.e., fat content) and to present values only influenced by fermentation and roasting (Figure 2). Appendix A show normalized values for powders and beans. Appendix A shows 2-way ANOVA results for monomers and PCs. As expected based on Folin–Ciocalteu and DMAC, UF/UR, and UF/CR powders had the highest concentrations of individual PCs DP 1–10 (Figure 2) and HF/HR powder had the lowest levels. However, HF/HR had the highest mDP of all treatments, at approximately 10, when calculated only factoring in oligomeric and polymeric PCs (Figure 1E), but this increase was not seen for mDP, including monomers present before thiolysis.

### 3.3. Quantification of Maillard Reaction Products 

The MRPs were quantified in LMW and HMW fractions and unfractionated extracts (Figure 3). The results for early MRPs did not show clear trends. Intermediate and late MRPs generally increased as expected from UR to CR to HR within all of the treatments. Fermentation had little impact on MRPs, as expected.

### 3.4. α-Glucosidase Enzyme Inhibition 

Figure 4 shows the curves from four-parameter log-logistic model fitting of α-glucosidase activity as a function of inhibitor concentration for each treatment. See Appendix A for raw (non-fitted) dose-response curves representing mean ± SEM values for each treatment. The extracts dose-dependently inhibited α-glucosidase activity. CF/CR was the most efficacious of all extracts and it was more potent than acarbose. At 250 µg/mL, all of the treatments but UF/HR and HF/HR inhibited enzyme activity greater than acarbose, and at concentrations ≥500 µg/mL, all the treatments were better inhibitors than acarbose (Figure 4). The extracts had the following IC_50_ values from most potent to least potent: 68.09 (CF/CR) > 115.15 (HF/UR) > 125.31 (UF/CR) > 134.27 (CF/UR) > 154.14 (UF/UR) > 158.33 (HF/HR) > 169.19 (UF/HR), with acarbose (A) exhibiting an IC_50_ value of 133.22 for comparison. Note that statistical comparisons of IC50 were not possible, since this is a parameter that is interpolated from a single curve for each treatment, fitted to multiple replicates at each dose. The curve parameters that were obtained for the four-parameter log-logistic model were also calculated for each treatment: Hill coefficient, minimum/maximum values, and EC_50_, so that the statistical significance of inhibition parameters could be determined (Table 2). EC_50_ trends generally mimicked those of IC_50_ values.

### 3.5. Identifying Predictors of α-Glucosidase Enzyme Inhibitory Activity

Simple linear regression was employed to determine whether strong linear relationships existed between the observed α-glucosidase inhibitory activities (IC_50_) and any of the individual chemical composition parameters of the cocoas and their extracts (Figure 5), as a means of tentatively identifying the composition factors that may influence inhibitory activities of cocoa. A strong negative correlation (*R* = −0.882) with a statistically significant non-zero slope was observed between the overall mDP calculated when factoring in monomers originally present in the sample, and IC_50_. This demonstrates that the enzyme activity decreases as overall mDP increases (i.e. inhibition increases). No other composition value (total polyphenols, total flavanols, individual PCs, etc.) was strongly correlated to α-glucosidase inhibitory activity or had a non-zero slope. However, moderate positive correlations with IC_50_ were observed for the early and intermediate MRPs from the HMW extract fraction (*R* ≥ 0.52 for both), demonstrating that the enzyme activity also increases as the concentration of these very large and complex compounds increase (i.e. inhibition decreases), which suggests that these MRPs either interfere with enzyme inhibition, or are markers of the loss of compounds that inhibit enzyme activity. Weak negative correlations with IC_50_ were observed for total flavanols, total PCs measured by HILIC, PC dimers, tetramers, and nonamers and early MRPs (LMW fraction) (∣*R*∣> 0.2 for all). Finally, weak positive correlations with IC_50_ were observed for intermediate MRPs (LMW) and late MRPs (HMW). 

## 4. Discussion

We generated seven different cocoa powders representing a broad range of possible fermentation and roasting conditions, and then assessed compositional and bioactivity differences of these powders while using a partial factorial design through controlled fermentation and further processing. This novel approach allowed for us to solely attribute differences to processing, since all of the products were produced from the same starting beans. 

Cocoa fermentation, as mentioned previously, is reliant upon multiple highly variable factors, and thus fermentation conditions employed in cocoa production around the world vary tremendously. The reported heap fermentations appear to start uniformly around pH 3.6, but our model system started slightly out of these ranges [27,28,29,38,39,40]. Our cool and hot fermentations both concluded under more acidic conditions than is typically reported. Beans of Trinitario and Forastero varieties typically range in initial pH from 6.3–6.8, decreasing to approximately 5.0–6.0 by the end of fermentation, whereas Criollo beans, while not as widely produced or studied as the Trinitario and Forastero varieties, have been noted for their characteristically low pH [39,40,41]. The high initial DO values can be attributed to the fresh mixing of bean and simulated pulp media, dropping significantly after 24 h and remaining ≤1 mg/L for the remainder of the fermentation. This model system fermentation is liquid based and it was designed to mimic typical conditions in the center of a well-mixed cocoa heap. The FI and cut test are traditional quality controls conducted on farm to assess the post-fermentation bean quality. These methods are based upon anthocyanin degradation and aglycone release throughout fermentation. The FI of raw unfermented beans typically range from 0.3–0.6 and increase to 1.3–1.4 during fermentation, with a value of ≥1 indicating the near-complete anthocyanin degradation and, thus, adequate and complete fermentation [2,42,43]. However, our FI values started at >1 (Figure 1B). These data, along with the acidic values for both pulp and bean in CF and HF systems, lead us to believe that these beans are of Criollo or Nacional variety, but this statement cannot be verified and is thus a limitation of this study. Criollo beans have low levels of anthocyanins when compared to other varieties, like Trinitario and Forastero, and they have a naturally low pH, which possibly explains the high acidity in our fermentation systems and inconclusive FI and cut test results [41]. Nacional beans grow in Ecuador and they are very similar to Criollo beans [41]. Traceability is often limited or not possible in a global commodity supply chain, such as that of cocoa beans, and thus the exact variety of the beans used in this study is unknown. 

The total polyphenol and flavanol content differed between powder and bean products, with powders having higher total concentrations of both polyphenols and flavanols due to concentration of flavanols in powders via the removal of flavanol-free cocoa butter during pressing. Overall, our cocoa composition data align with previously published reports that fermentation and roasting significantly reduce native flavanol levels in cocoa beans [5,11,14,17,32,44,45,46]. We assessed the interaction of both steps by evaluating them in varying combinations, while most previous studies have investigated the implications of fermentation and roasting independently of each other. Roasting is often considered to be the key phase in cocoa processing in terms of defining the sensory characteristics of a finished product by producing characteristic aromas, flavors, and texture of beans. Yet, roasting is an extension of the flavanol reduction that begins during fermentation. The epimerization and polymerization of flavanols, as well as reactions with larger structures, such as proteins, polysaccharides, and MRPs, can all influence the flavonoid levels in processed cocoa [5,46]. This is clearly demonstrated by our data across a broad range of polyphenol and flavanol assays. 

Significantly lower levels of all measured compounds in the most harshly processed cocoa, HF/HR, as compared to other treatments, demonstrate that prolonged high temperature exposure via hot fermentation, followed by hot roasting degrades native polyphenols and flavanols. However, the high mDP that was reported for both the HF/UR and HF/HR suggests that fermentation might have a larger influence on polymerization of larger molecular weight compounds than originally thought, or that the interaction between fermentation and roasting at high temperatures is crucial in the development of large PCs. This agrees with studies suggesting that levels of high molecular weight PCs increase with greater roasting time and temperatures [10,17]. 

Flavanols, particularly HMW flavanols, generally have poor intestinal absorption, and we chose to examine the inhibition of a digestive enzyme, such as α-glucosidase (a brush-border enzyme), in order to examine a mechanism located where these compounds are present at their highest concentrations in vivo (the lumen or epithelium of the intestines). We also chose α-glucosidase based on data suggesting that cocoa exerts greater inhibition on this enzyme than on the pancreatic α-amylase and lipase [13]. Through the inhibition of α-glucosidase, cocoa compounds have the potential to slow down carbohydrate digestion and post-prandial absorption of glucose, which blunts blood glucose excursions. Extracts that were produced in this study appear to be effective dietary inhibitors of α-glucosidase, most notably the CF/CR treatment, with an IC_50_ of 68.1 µg/mL (~50% lower than that of acarbose). The level of acarbose that is typically present in the gut is between 25–100 µg/mL when taken as recommended, which is within the concentrations used in the present study, and thus represents a relevant control for this assay. At an IC_50_ of 68.1 µg/mL, CF/CR falls within the typical acarbose range, but it is two-fold more effective in inhibiting 50% of enzymatic activity. The CF/CR processing parameters that were applied in this study were within ranges used in industrial cocoa powder production and have promising potential to surpass the activity of acarbose in vivo. In addition to CF/CR, HF/UR, and UF/CR also had lower IC_50_ values than acarbose. Furthermore, all of the cocoas were more effective than acarbose at higher doses, such as 500 µg/mL (Figure 4). While 500 µg/mL is approximately five-fold higher than typical acarbose concentrations within the gut, it is still a highly relevant dietary dose at 6.67 g of original cocoa product or approximately 0.24 square of baking chocolate. Even at 250 μg/mL, all the cocoas treatments, except UF/HR and HF/HR, had better inhibitory effects than acarbose, and these were treatments that were not significantly different than acarbose (Figure 4). While it is promising that select processing conditions improve IC_50_ values, it is noteworthy that even the harshest of conditions have similar inhibitory activity as compared to UF/UR (no fermentation or roasting processing). The least-processed cocoa (UF/UR) was not the best inhibitor nor was the most harshly-processed cocoa (HF/HR) far worse than all other treatments, which suggests that, in this particular instance, processing, at worst, does not negatively affect activity and, at best, can actually greatly enhance inhibitory activity. 

The Hill coefficient is the slope of the curve at the inflection point and it can be used to interpret the binding behavior and kinetic activity of the inhibitor’s target. Furthermore, a Hill coefficient of 1 is considered to be standard and is generally understood to have a single inhibitor binding site or a simple kinetic mechanism (Table 2). With a steeper curve, the Hill coefficient increases, which indicated that enzyme inhibition increases with an overall decrease in concentration range. Although the reasons for these steep dose-response curves are poorly understood, several mechanisms can potentially explain this action, including an increased number of inhibitor sites, an inhibitor undergoing a phase transition with increased concentration, as well as when the enzyme concentration exceeds the equilibrium constant for the inhibitor. With the exception of UF/CR and CF/CR, all of the treatments have standard slopes and, therefore, can be considered to have simple kinetic mechanisms. EC_50_, or the half-maximal effective concentration in relation to the control (i.e. acarbose), can be used in combination with log-logistic parameters to estimate the IC_50_ of each treatment. There are slight differences between these and IC_50_ values due to dependence on the control for EC_50_ values. Yet, CF/CR is the most powerful inhibitor of α-glucosidase with both EC_50_ and IC_50_ values being lower than all other treatments, including acarbose. Overall, our data do not support the hypothesis that an inverse relationship exists between the processing intensity and α-glucosidase inhibitory activity of cocoa powder. 

Once we determined the α-glucosidase inhibitory activities of these cocoas, we then wished to determine whether measured concentrations of putative bioactive cocoa compounds were associated with inhibitory activity. We employed a simple linear regression approach to achieve this, similar to our previous studies [13,47] (Figure 5). Of all compositional measures, the only strong correlation seen was between IC_50_ and the overall flavanol mDP, with decreasing IC_50_ as mDP increased. The lack of strong correlations for most measures in Figure 5 aligns with previous work demonstrating that, although processing induced significant losses in total polyphenols and total flavanols, these compositional changes did not uniformly influence bioactivity, but, rather, increasing mDP had a stronger influence on cocoa bioactivity [13,25,48,49]. This finding provides evidence suggesting that cocoa processing could be specifically tailored to promote flavanol polymerization as a means to enhance α-glucosidase activity. Additionally, our data begin to suggest that MRPs may be useful as markers of activity in cocoa. These compounds are intriguing markers, as they may directly contribute to α-glucosidase inhibition due to structural similarities with their carbohydrate precursors, and they are also sensitive indicators of processes, such as roasting. The quantification of these compounds is complex, with limited understanding of structure and activity. Our results preliminarily suggest that longer and higher roasting times/temperatures, which often result in the increased production of these MRPs, negatively impact the α-glucosidase inhibitory activity of the extract, but further investigation into these compounds is needed to fully elucidate the impact that they have on specific digestive enzymes. Finally, our data suggest that traditional putative markers, such as total polyphenols and total or specific flavanols, may not be sufficient for predicting α-glucosidase inhibitory activity of cocoa. Whether this applies to other bioactivities, in vitro and in vivo, remains to be seen.

The main novelties of this study were in (1) our use of a single uniform batch of raw cocoa beans as the starting material for all the processing treatments, (2) coupled with examining variations in fermentation and roasting in combination as opposed to separately, and (3) challenging the hypothesis that traditional putative bioactives in cocoa are correlated with a given bioactivity. Previous studies have often relied on information that was reported through various levels of the cocoa supply chain, often resulting in unknowns regarding origin of the beans, processing conditions, as most cocoa production processes in country of origin, especially fermentation, lack robust controls or recordkeeping, which results in considerable variation and poor traceability. Our controlled system eliminates many of these external challenges, providing confidence that differences between the cocoa powders were due to the fermentation and roasting treatments, rather than unknown factors. While the model fermentation system that was used in our study was not designed to physically mimic the conditions found in on-farm cocoa fermentation, fermentation of cocoa beans while using this system results in biochemical changes analogous to those reported in on farm fermentation. Conducting the fermentation step in this model system allowed for us to produce cocoa powders with acceptable chemical composition, which were subject to known and controlled fermentation conditions. We are currently expanding on this research by investigating each cocoa powder’s ability to prevent obesity-induced GI and systemic inflammation and gut barrier dysfunction in vivo.

This study is not without limitations. Quantifying individual PCs ranging from DP 1–10, although expanding beyond the monomeric flavan-3-ols often focused on in cocoa products, still leaves many large molecular weight compounds yet to be individually quantified. Additionally, α-glucosidase inhibition is just one specific mode of bioactivity that we chose to focus on, based on previous reports. Although our results begin to expand on this powerful inhibitory effect, the exact compositional factors that are involved remain to be elucidated. Our processing parameters could be expanded to include a wider range of processing conditions to address these limitations moving forward, to further optimize the α-glucosidase inhibitory activity. Additional work is also needed to study the finer variations of the optimal parameters that we identified here to further enhance the inhibitory activity. This expansion of treatment conditions tested, as well as further the fractionation of powder extracts to identify specific components that are associated with α-glucosidase inhibitory activity, would allow for a more comprehensive understanding of the impact that each processing step has on various compositional factors and bioactivity of dietary cocoa. By extending characterization to lignins and the speciation of melanoidins and other MRPs, further evidence could be provided to explain the mechanisms that govern enzymatic α-glucosidase inhibition by cocoa. This in vitro work also needs to be extended in vivo to further reinforce and clarify the mechanisms behind cocoa’s enzymatic inhibition influence in both animals and humans, including, but not limited to, maltose versus glucose tolerance tests with cocoa consumption, and long-term effects on diet-induced obesity and glucose intolerance. Furthermore, yeast-derived α-glucosidase is a good, inexpensive starting point to screen for α-glucosidase inhibitory activity *in vitro*. Although yeast derived α-glucosidase is not completely identical to mammalian α-glucosidase, it is commonly selected for anti-diabetic investigations and it can be used for preliminary screening before investigating further in costly mammalian (i.e. rat acetone intestinal powder or Caco-2 cells) or human recombinant enzymes. For our purposes, yeast α-glucosidase was initially used to demonstrate the concept that processing does not necessarily eliminate cocoa’s α-glucosidase inhibitory activity and, therefore, was not intended to be a definitive test of in vivo human relevance. Cocoa, as well as other polyphenol-rich substances, have exhibited powerful inhibitory activity when using mammalian α-glucosidase, showing promise for our data moving forward [16,50,51,52]. However, further studies must be conducted to establish relevance between these models and the human digestive tract, as well as specific mechanisms of action. 

## 5. Conclusions 

Overall, this study demonstrates that processed cocoa powders are promising inhibitors of α-glucosidase, despite a significant reduction in native flavanol composition during fermentation and roasting, and fermentation and roasting can improve inhibitory activity when compared to raw cocoa. We report the novel finding that cocoa processing might generate compounds with α-glucosidase inhibitory activity, and that non-traditional markers, such as MRPs, may be more informative than traditional markers, such as total and individual polyphenols and flavanols. These observations support our hypothesis that reductions in native polyphenols and flavanols do not necessarily dictate a reduction in activity and, furthermore, that products of fermentation and roasting do, in fact, contribute to cocoa bioactivity. Further investigation is needed to determine the identity of compounds that explain this activity. Finally, it remains to be seen whether these findings apply to other bioactivities of cocoa, but the present study provides the proof of concept needed to justify such investigations.

## Figures and Tables

**Figure 1 antioxidants-08-00635-f001:**
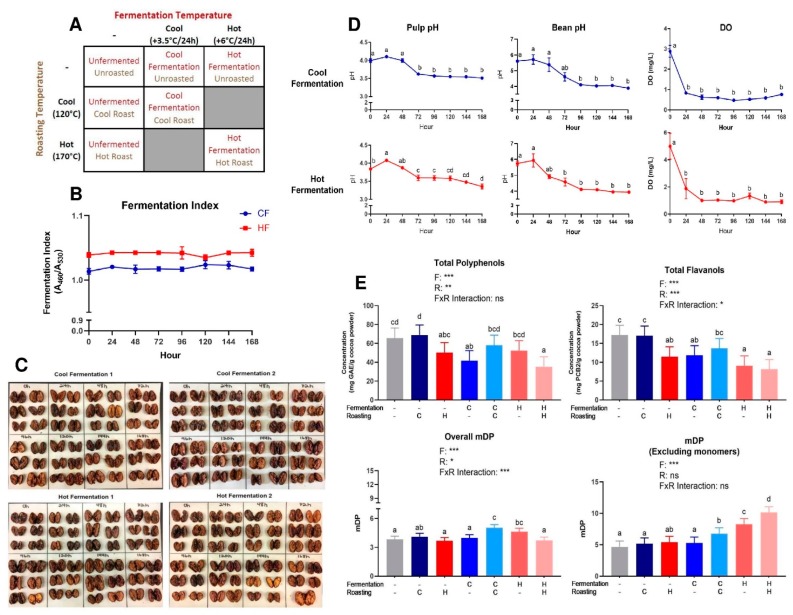
(**A**) Cocoa processing model system to evaluate the impact of combined fermentation and roasting parameters, producing seven total cocoa powders: unfermented/unroasted (UF/UR), unfermented/cool roast (UF/CR), unfermented/hot roast (UF/HR), cool fermentation/unroasted (CF/UR), cool fermentation/cool roast (CF/CR), hot fermentation/unroasted (HF/UR), and hot fermentation/hot roast (HF/HR). (**B**) Fermentation index as a ratio of absorbance at 460 nm:530 nm, with values ≥1 indicating a complete fermentation. Note broken axes for ease of interpretation on select graphs. Values are presented as the mean ± SEM of fermentation replicates within treatments. Significant between time points for each value was determined by one-way ANOVA and Tukey’s HSD post-hoc test (*p* < 0.05). Time points with different letters are significantly different within values. (**C**) Cut test for all fermentations performed. (**D**) pH of simulated pulp media/bean nib and dissolved oxygen (DO). It is important to note that for bean nib measurements, these values do not quantify the pH of the cocoa bean itself, but rather the acidity derived from bean acids diluted in water. These nib values are useful for comparison between the pH of the solution produced by beans at different time points. (**E**) Total polyphenols in each cocoa powder, expressed in gallic acid equivalents. Total flavanols from cocoa powder expressed in procyanidin B2 equivalents. Overall mean flavanol degree of polymerization (mDP) for the total flavanols in cocoa powder- native monomers were accounted for in calculation. Mean flavanol degree of polymerization for oligomers and polymers in cocoa powder (not including native monomers); Note broken axes for ease of interpretation. All values are presented as the least squares (LS) means with upper and lower confidence interval (CI). Significance between treatments was determined by two-way ANOVA for the roasting and fermentation temperature effects using type III sums of squares to account for unbalanced data, followed by post-hoc comparisons of LSMEANS (*p* < 0.05). Normality was checked for each variable visually and with the Shapiro–Wilks test, and if needed, transformed (Log or Box–Cox) prior to running the ANOVA and post-hoc test. Treatments with different superscript letters (a–d) in Figure 1D and E are significantly different within values. Legends above graphs indicate treatment (F = fermentation; R = roasting) main effect and interactions as determined by two-way ANOVA, * *p* < 0.05, ** *p* < 0,01, *** *p* < 0.001.

**Figure 2 antioxidants-08-00635-f002:**
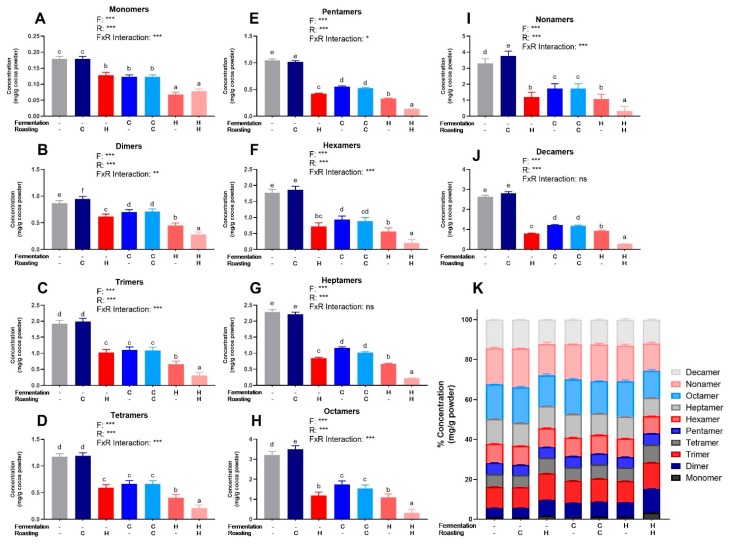
(**A–J**) Levels of individual procyanidin compounds in cocoa powders as measured by HILIC UPLC-MS/MS. (**K**) Individual values as a% of total measured procyanidins for each treatment. Values are normalized to the fat-free mass of each treatment to account to varying fat content. All values are presented as the LS means with upper and lower CI. Significance between treatments was determined by two-way ANOVA for the roasting and fermentation temperature effects using type III sums of squares to account for unbalanced data, followed by post-hoc comparisons of LSMEANS (*p* < 0.05). Normality was checked for each variable visually and with the Shapiro–Wilks test, and if needed, transformed (Log or Box-Cox) prior to running the ANOVA and post-hoc test. Treatments with different superscript letters (a–e) are significantly different within values. Legends above graphs indicate treatment (F= fermentation; R= roasting) main effect and interactions as determined by two-way ANOVA, * *p* < 0.05, ** *p* < 0,01, *** *p* < 0.001, refer to Appendix A for numerical values.

**Figure 3 antioxidants-08-00635-f003:**
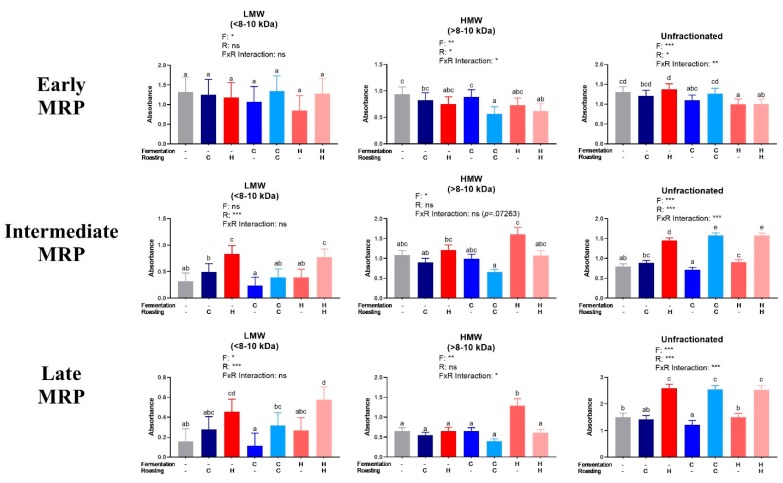
Analysis of low molecular weight (LMW) and high molecular weight (HMW) cocoa extract fractions and the unfractionated extract for early, intermediate, and late Maillard reaction products (MRP). Early MRP were quantified at 0.15625 mg/mL at 280 nm, intermediate MRP were quantified at 2.5 mg/mL at 360 nm, and late MRP were quantified at 5 mg/mL (LMW, <8–10 kDa), 2.5 mg/mL (HMW, >8–10 kDa), and 10 mg/mL (unfractionated extract) at 420 nm. Note that absolute absorbances are reported due to lack of adequate standards. Each bar represents the LS means with upper and lower CI. Significance between treatments was determined by two-way ANOVA for the roasting and fermentation temperature effects using type III sums of squares to account for unbalanced data, followed by post-hoc comparisons of LSMEANS (*p* < 0.05). Normality was checked for each variable visually and with the Shapiro–Wilks test, and, if needed, transformed (Log or Box-Cox) prior to running the ANOVA and post-hoc test. Treatments with different superscript letters (a–e) are significantly different within values. Legends above graphs indicate treatment (F= fermentation; R= roasting) main effect and interactions as determined by two-way ANOVA, **p* < 0.05, ***p* < 0,01, ****p* < 0.001, refer to Appendix A for numerical values.

**Figure 4 antioxidants-08-00635-f004:**
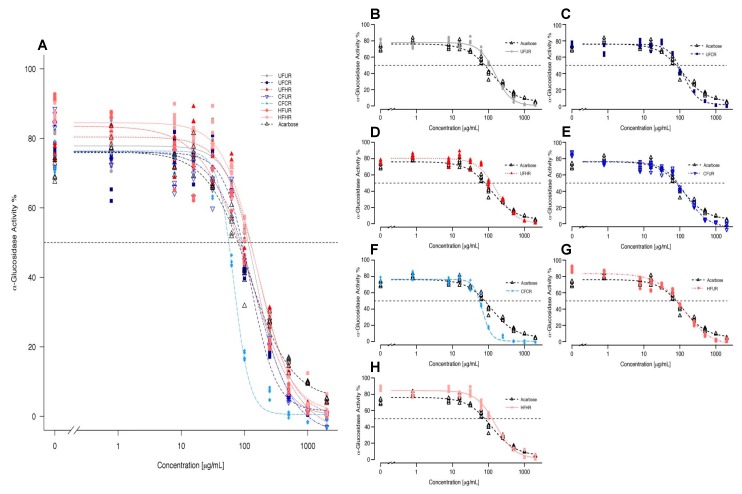
Overlay fitted curves obtained for α-glucosidase activity (% activity compared to no inhibitor) via 4-parameter log-logistic model for all cocoa powder extracts (**A**) and treatments plotted individually against acarbose (**B**–**H**). Individual points represent inhibition values for individual replicates at each concentration. Dotted line represents IC_50_ values. Refer to Table 2 for numerical values.

**Figure 5 antioxidants-08-00635-f005:**
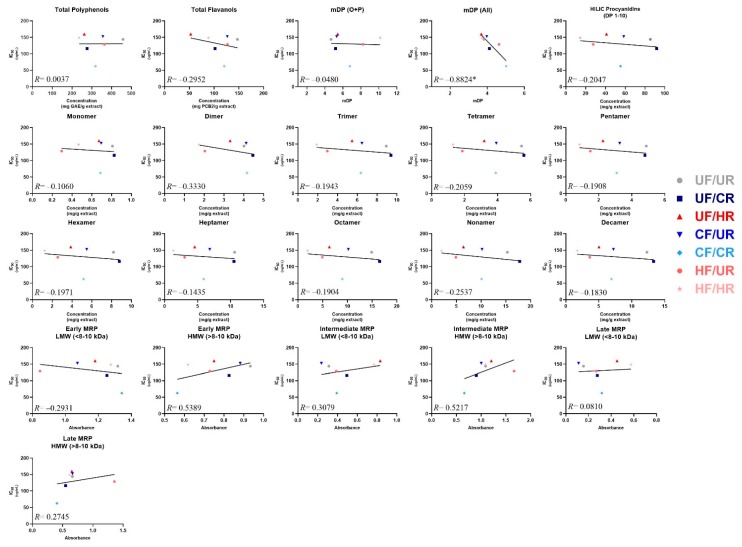
Correlations between cocoa powder extract composition and enzyme IC_50_ values. For mean degree of polymerization (mDP), O + P: oligomers + polymers (not factoring in monomers present prior to thiolysis), and All: (including monomers present prior to thiolysis). Note the x-axis for mDP graphs have a minimum value of 1 mDP because there cannot be an mDP value of <1. Note that early, intermediate, and late MRP are presented as absolute absorbance. Individual points represent mean composition values and calculated IC_50_ values for each treatment. Lines represent the least-squares regression line for each plot. Note that composition values are for the cocoa extract, not powders, since the extract was evaluated for enzyme activity. *indicates that the slope is significantly different from zero.

**Table 1 antioxidants-08-00635-t001:** Composition of simulated pulp media.

Reagent *^e^*	Mass *^a^* (g)
Simulated Pulp Solution *^b^*	
Citric acid	40
Yeast extract	20
Peptone	20
Calcium lactate pentahydrate	4
Tween 80	4
Magnesium-Manganese Solution *^c^*	
Magnesium sulfate-heptahydride	2
Manganese sulfate-monohydride	0.8
Sugar Solutions *^d^*	
Sucrose	100
Glucose	160
Fructose	180

*^a^* per four liters of solution; *^b^* reagents were combined with 1600 mL DI water, pH adjusted to 3.6 using 1 N NaOH, and adjusted to a final volume of 2.4 L DI water before autoclaving; *^c^* reagents were combined with 400 mL of DI water; *^d^* reagents were made separately into 400 mL solutions; *^e^* solutions c and d were combined with autoclaved solution b to begin fermentation.

**Table 2 antioxidants-08-00635-t002:** Enzyme inhibition as analyzed by four-parameter log-logistic model. Significance indicated by **p* < 0.05, ***p* < 0.01, ****p* < 0.001.

Parameter	Treatment			Estimate	Std. Error	LCL (2.5%)	UCL (97.5%)	*t*-Value	*p-*Value	Significance
	F	R	Abbreviation							
**Hill Coefficient**	-	-	(UF/UR)	1.9696	0.141725	1.691	2.248	13.8973	<2.2 × 10^−16^	***
	-	Cool	(UF/CR)	2.142171	0.177728	1.793	2.491	12.0531	<2.2 × 10^−16^	***
	-	Hot	(UF/HR)	1.603793	0.113242	1.381	1.826	14.1626	<2.2 × 10^−16^	***
	Cool	-	(CF/UR)	1.429009	0.129739	1.174	1.684	11.0145	<2.2 × 10^−16^	***
	Cool	Cool	(CF/CR)	3.465039	0.425681	2.629	4.301	8.14	3.17 × 10^−16^	***
	Hot	-	(HF/UR)	1.142063	0.107241	0.931	1.353	10.6495	<2.2 × 10^−16^	***
	Hot	Hot	(HF/HR)	1.627053	0.113836	1.403	1.851	14.2929	<2.2 × 10^−16^	***
	Acarbose		(A)	1.30504	0.113057	1.083	1.527	11.5432	<2.2 × 10^−16^	***

**Minimum Value**	-	-	(UF/UR)	0.378196	1.496104	−2.561	3.318	0.2528	0.80054	
	-	Cool	(UF/CR)	1.513805	1.388439	−1.214	4.242	1.0903	0.27611	
	-	Hot	(UF/HR)	−0.030323	1.857523	−3.68	3.619	−0.0163	0.98698	
	Cool	-	(CF/UR)	−4.888001	2.078745	−8.972	−0.804	−2.3514	0.01909	*
	Cool	Cool	(CF/CR)	0.516432	0.977596	−1.404	2.437	0.5283	0.59755	
	Hot	-	(HF/UR)	−5.208691	2.488998	−10.1	−0.318	−2.0927	0.03689	*
	Hot	Hot	(HF/HR)	1.208122	1.755526	−2.241	4.657	0.6882	0.49166	
	Acarbose		(A)	4.867636	2.043044	0.854	8.882	2.3825	0.01757	*

**Maximum Value**	-	-	(UF/UR)	77.81598	0.894435	76.059	79.573	87.0002	<2.2 × 10^−16^	***
	-	Cool	(UF/CR)	75.880224	0.889772	74.132	77.628	85.2806	<2.2 × 10^−16^	***
	-	Hot	(UF/HR)	80.391462	0.940686	78.543	82.24	85.4604	<2.2 × 10^−16^	***
	Cool	-	(CF/UR)	76.297669	1.107626	74.121	78.474	68.8839	<2.2 × 10^−16^	***
	Cool	Cool	(CF/CR)	76.360965	0.968329	74.458	78.264	78.8585	<2.2 × 10^−16^	***
	Hot	-	(HF/UR)	83.518081	1.382998	80.801	86.235	60.3892	<2.2 × 10^−16^	***
	Hot	Hot	(HF/HR)	84.494321	0.96061	82.607	86.382	87.959	<2.2 × 10^−16^	***
	Acarbose		(A)	76.123584	1.073488	74.014	78.233	70.9123	<2.2 × 10^−16^	***

**EC_50_**	-	-	(UF/UR)	153.38225	8.067667	137.53	169.23	19.012	<2.2 × 10^−16^	***
	-	Cool	(UF/CR)	122.95011	6.486545	110.21	135.7	18.9546	<2.2 × 10^−16^	***
	-	Hot	(UF/HR)	169.26537	10.618809	148.4	190.13	15.9401	<2.2 × 10^−16^	***
	Cool	-	(CF/UR)	146.08776	9.519332	127.38	164.79	15.3464	<2.2 × 10^−16^	***
	Cool	Cool	(CF/CR)	67.824539	1.888888	64.113	71.536	35.9071	<2.2 × 10^−16^	***
	Hot	-	(HF/UR)	127.62786	9.183673	109.58	145.67	13.8973	<2.2 × 10^−16^	***
	Hot	Hot	(HF/HR)	155.5339	8.935954	137.98	173.09	17.4054	<2.2 × 10^−16^	***
	Acarbose		(A)	119.95973	10.225587	99.869	140.05	11.7313	<2.2 × 10^−16^	***

**IC_50_*^a^***	-	-	(UF/UR)	154.1448	-	132.33 *^a^*	175.92 *^b^*	-	-	-
	-	Cool	(UF/CR)	125.30945	-	108.24 *^a^*	142.15 *^b^*	-	-	-
	-	Hot	(UF/HR)	169.1858	-	139.08 *^a^*	199.97 *^b^*	-	-	-
	Cool	-	(CF/UR)	134.26837	-	105.91 *^a^*	162.82 *^b^*	-	-	-
	Cool	Cool	(CF/CR)	68.09163	-	63.22 *^a^*	72.61 *^b^*	-	-	-
	Hot	-	(HF/UR)	115.14538	-	86.2 4 *^a^*	144.88 *^b^*	-	-	-
	Hot	Hot	(HF/HR)	158.33224	-	132.88 *^a^*	184.1 *^b^*	-	-	-
	Acarbose		(A)	133.22093	-	102.04 *^a^*	165.78 *^b^*	-	-	-

*^a^* IC_50_ LCL at 5%; *^b^* IC_50_ UCL at 95%.

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
