# Peer review of "Flavanol Polymerization Is a Superior Predictor of α-Glucosidase Inhibitory Activity Compared to Flavanol or Total Polyphenol Concentrations in Cocoas Prepared by Variations in Controlled Fermentation and Roasting of the Same Raw Cocoa Beans"

_antioxidants, 2019, doi:10.3390/antiox8120635_

Round 1

Reviewer 1 Report

This manuscript written by Kathryn C. Racine and coworkers described the α-glucosidase inhibitory activity and nutrient of cocoa beans. The authors spent a lot of efforts to control the producing process of cocoa beans and measured the polyphenols, melanoidins, and in vitro α-glucosidase inhibition effects. I believe this manuscript can be published in Antioxidants after some revision.

The comments are list below:

1. The α-glucosidase is related to diabetes, what is the relationship between α-glucosidase and antioxidents?

2. The authors mimicked several processes and evaluate the α-glucosidase inhibitory activity of each process. I think the author should do further experiments to tell us an exact method (cool fermentation/cool roast for how long) which provides the maximum α-glucosidase inhibition effects.

3. Too many lines in figure 4, I think the authors can delete half less effective lines.

Author Response

This manuscript written by Kathryn C. Racine and coworkers described the α-glucosidase inhibitory activity and nutrient of cocoa beans. The authors spent a lot of efforts to control the producing process of cocoa beans and measured the polyphenols, melanoidins, and in vitro α-glucosidase inhibition effects. I believe this manuscript can be published in Antioxidants after some revision.

The comments are listed below:

The α-glucosidase is related to diabetes, what is the relationship between α-glucosidase and antioxidants?

We thank the reviewer for bringing this question to our attention. Although inhibition of α-glucosidase is not an antioxidant activity, cocoa flavanols are antioxidants with numerous reported health benefits, and we believe is within the scope of this journal (specifically, the scope includes “natural or synthetic antioxidants and their relevance to health and disease”). Furthermore, we feel this article is a good fit for the “Antioxidants in cocoa” special issue. We have added some detail to the introduction in order to clarify this important connection, included in Lines 48-51:

“These flavanols exhibit potent antioxidant and health-protective activities, including the modulation of oxidative stress and potentially reducing the risk of various chronic conditions, such as cardiovascular disease (CVD), type II diabetes mellitus (T2D), and different forms of cancer.”

The authors mimicked several processes and evaluate the α-glucosidase inhibitory activity of each process. I think the author should do further experiments to tell us an exact method (cool fermentation/cool roast for how long) which provides the maximum α-glucosidase inhibition effects.

We agree with the reviewer on this point and intend to continue our work to further refine the cocoa processing conditions to produce a product with maximum α-glucosidase inhibition effects. Given the large amount of data we have for the present paper, we feel the present data are sufficient but we agree that this is important for future studies. We added the following text on Lines 568-570 to address this:

“Additional work is also needed to study finer variations of the optimal parameters we identified here in order to further enhance inhibitory activity.”

Too many lines in Figure 4. I think the authors can delete half less effective lines.

We appreciate the reviewer pointing this out. To accommodate, we’ve modified Figure 4 to include not only the combined plot (as in the original figure) but also individual panels that represent each treatment plotted against the control, acarbose. This should make Figure 4 much easier to interpret.

Reviewer 2 Report

Target of this studies was to find effects of industrial processes on composition of cocoa beans.

Cocoa beans undergo many processes leading to the loss of their ingredients as polyphenols and others. To disclose influence of these processes the authors have been carried out appropriate experiments concerning fermentation model system and cocoa processing, quantifications of polyphenols, Maillard reaction products, evaluation of α-glucosidase enzyme inhibition, identification of possible markers of α-glucosidase inhibition.

The studies presented in this manuscript show that the processes accompanying the processing of cocoa beans may favor to increase the bioactivity and bioavailability of this nutritionally valuable food product. What's more, by-products that form during the processing of cocoa beans can contribute to cocoa bioactivity. The authors finding indicate that Maillard reaction products may be more informative markers than traditional ones such as contents of total and individual polyphenols and flavanols.

The results of experiments indicate that conducting further research in the field of identification of new compounds and their mechanisms of action will allow rational health assessment of cocoa beans.

I suggest only a minor revision in order to correct several misspelling e.g. Maillard “reactio”

Author Response

Target of this studies was to find effects of industrial processes on composition of cocoa beans.

Cocoa beans undergo many processes leading to the loss of their ingredients as polyphenols and others. To disclose influence of these processes the authors have been carried out appropriate experiments concerning fermentation model system and cocoa processing, quantifications of polyphenols, Maillard reaction products, evaluation of α-glucosidase enzyme inhibition, identification of possible markers of α-glucosidase inhibition.

The studies presented in this manuscript show that the processes accompanying the processing of cocoa beans may favor to increase the bioactivity and bioavailability of this nutritionally valuable food product. What's more, by-products that form during the processing of cocoa beans can contribute to cocoa bioactivity. The authors finding indicate that Maillard reaction products may be more informative markers than traditional ones such as contents of total and individual polyphenols and flavanols.

The results of the experiments indicate that conducting further research in the field of identification of new compounds and their mechanisms of action will allow rational health assessment of cocoa beans.

I suggest only a minor revision in order to correct several misspelling e.g. Maillard “reactio”

We thank the reviewer for pointing out these mistakes. We have made the following corrections but please let us know if there are any additional mistakes to be fixed:

Line 147: “…rehydrated beans (60 kg) were mixed with…”

Line 374: “ Finally, weak positive correlations…”

Line 375: “…MRPs…”

Line 436: “Extracts produced in this study…”

Line 439: “…within the concentrations used in this present study…”

Line 449: “…it is noteworthy that even the harshest…”

Line 484: “…inhibitory activity of the extract…”

Reviewer 3 Report

The authors in this study demonstrates that processed cocoa powders are promising inhibitors of α-glucosidase, despite a significant reduction in native flavanol composition during fermentation. The manuscript is very interesting and is well written. It is a very good presented and the conclusion supported by results. I suggest that  the authors can reporting the importance ofpolyphenols  for human health in the introduction section. 

Author Response

The authors in this study demonstrates that processed cocoa powders are promising inhibitors of α-glucosidase, despite a significant reduction in native flavanol composition during fermentation. The manuscript is very interesting and is well written. It is a good presented and the conclusion supported by results. I suggest that the authors can reporting the importance of polyphenols for human health in the introduction section.

We thank the reviewer for their comment and have addressed this issue in the introduction sentences.

Line 44-45: “Cocoa beans (Theobroma cacao) are concentrated dietary sources of flavanols, a subclass of polyphenols, which are thought to be responsible for many bioactivities of cocoa.”

Line 48-51: “These flavanols exhibit potent antioxidant and health-protective activities, including the modulation of oxidative stress and potentially reducing the risk of various chronic conditions, such as cardiovascular disease (CVD), type II diabetes mellitus (T2D), and different forms of cancer.”

Reviewer 4 Report

The manuscript reported a very interesting and articulated research about the a-glucosidase activity inhibition by some cocoa constituents.

Some small observations are reported below.

The text have to be check. Some typing mistakes occur along it. For example at line 132 and 421, the word “were” was repeated or, in other parts, capital letter was not use. Authors should carefully verify all text.

As the authors reported and exhaustively discussed the variety of cocoa is important to define some characters. The research is carried out with unidentified commercial cocoa beans. The authors assumed the cocoa beans were Criolllo or National variety on the characters observation. This could be not sufficient from a scientific point of view. I agree with authors when they established that the traceability and certainty of variety are very difficult to assess.

Introduction part.

Lines 49-51. Authors reported some literature references about some cocoa characters. The reference 14 should be check, it is about other product. They  also should see  the observation at References paragraph.

Material and methods

2.2.1 Fermentation

Even if the authors used a fermentation method reported in another research (citations 31 and 32), they should describe what they mean as “simulated pulp media” (its composition mainly) at lines 133-134.

In the same paragraph, at lines 144-145, authors should report the time applied to dehydrate beans. They could report it approximately if they have not the precise one. This time at temperature as that applied could promote Maillard derived compounds.

2.4 Polyphenol extract and characterization

At lines 184-185 authors reported “Cocoa monomeric flavanols and PCs (DP 1-10) were quantified by HILIC UPLC-MS/MS as previously described”. It is not clear at which point they “previously” described it. If they refer to Racine et al [36], I suggest changing text into “…as previously described by….

2.5 Melanoidin dialysis

Surely, the extract used was the phenol extract reported in the previous paragraph, but for a better comprehension of text, authors should define it (line 189).

2.7 Safety consideration

Why authors reported this paragraph, if it is not applicable?

Results

3.1. Fermentation model system and cocoa processing

Lines 262-263. Authors should better describe what they mean as “300+” sample size. No description of this was find in “material an methods” part neither in supplementary one.

3.2. Characterization and quantification of polyphenols

I did not find figure 1E about polyphenol and flavanols content of powder. Without this figure (and data) it is impossible verify discussion.  Authors should report it.

Discussion

Line 382. Authors should explain what they mean as “genetic composition”. From a lexical point of view, the terminology appears quite strange.

References paragraph.

The authors should verify the references n.14, then, it is correct reported in this manuscript they should carefully check. Di Mattia, C.; Sacchetti, G.; Seghetti, L.; Piva, A.; Mastrocola, D. “Vino Cotto” Composition and Antioxidant Activity as Affected by Non Enzymatic Browning. 2007, 19, 413–424. The name of journal is wrongly reported.

Author Response

The manuscript reported a very interesting and articulated research about the a-glucosidase activity inhibition by some cocoa constituents.

Some small observations are reported below.

The text have to be check. Some typing mistakes occur along it. For example at line 132 and 421, the word “were” was repeated, or in other parts, capital letter was not used. Authors should carefully verify all text.

We thank the reviewers for catching these mistakes and have carefully verified the text. Please see comments from Reviewer 2 for specific corrections.

As the authors reported and exhaustively discussed the variety of cocoa is important to define some characters. The research is carried out with unidentified commercial cocoa beans. The authors assumed the cocoa beans were Criollo or National variety on the characters observation. This could be not sufficient from a scientific point of view. I agree with authors when they established that the traceability and certainty of variety are very difficult to assess.

We appreciate the reviewer pointing this out. We do not assume anything, but rather wish to convey our data that suggest these are Criollo or Nacional beans. We have modified the text to state this as follows on Line 403-405:

“These data, along with the acidic values for both pulp and bean in CF and HF systems, lead us to believe that these beans are of Criollo or Nacional variety, but this statement cannot be verified and is thus a limitation of this study.”

Introduction part.

Line 49-51. Authors reported some literature references about some cocoa characters. The reference 14 should be check, it is about other product. They also should see the observation as References paragraph.

We appreciate the reviewer pointing this detail at. Reference 14 has been removed from the reference list. We apologize for the confusion.

Material and methods

2.2.1 Fermentation

Even if the authors used a fermentation method reported in another research (citations 31 and 32) they should describe what they mean as “simulated pulp media” (its composition mainly) at lines 133-134.

We appreciate the reviewer pointing this out. To better describe the simulated pulp media used for the fermentation, we have included Table 1: Composition of simulated pulp media (Line 151).

In the same paragraph, at lines 144-145, authors should report the time applied to dehydrate beans. They could report I approximately if they have not the precise one. This time at temperature as that applied could promote Maillard derived compounds.

We appreciate the reviewer’s suggestion. We have added the time applied to dehydrate the beans in Line 161:

“After 168 h of fermentation, the beans were drained to remove the remaining simulated pulp media, spread evenly onto baking sheets and oven dried at 65.5C for 24-26 h or until the moisture content fell below 8%.”

2.4 Polyphenol extract and characterization

At lines 184-185 authors reported “Cocoa monomeric flavanols and PCs (DP 1-10) were quantified by HILIC UPLC-MS/MS as previously described”. It is not clear at which point they “previously” described it. If they refer to Racine el al [36], I suggest changing text into “…as previously described by…”

We appreciate the reviewer’s comment and have clarified this statement in Line 204:

“Cocoa monomeric flavanols and PCs (DP 1-10) were quantified by HILIC UPLC-MS/MS based on the method of Racine et al.”.

2.5 Melanoidin dialysis

Surely, the extract used was the phenol extract reported in the previous paragraph, but for a better comprehension of text, authors should define it (line 189).

We appreciate the reviewer point this out. We have clarified Line 209:

“Polyphenol-rich extracts were re-dissolved in extraction solution…”.

2.7 Safety consideration

Why authors reported this paragraph, if it is not applicable?

We apologize for this oversite. This section has been removed.

Results

3.1 Fermentation model system and cocoa processing

Lines 262-263. Authors should better describe what they mean as “300+” sample size. No description of this was find in “materials and methods” part neither in supplementary one.

We appreciate the reviewer pointing this out. We have clarified this statement as follows in Line 287-289:

“Cut test results showed similar results as there was no true progression of color from purple to brown beans. However, the sample size used was much smaller than what is traditionally seen, with typical cut tests consisting of over 300 cut beans.”

3.2 Characterization and quantification of polyphenols

I did not find figure 1E about polyphenol and flavanols content of powder. Without this figure (and data) it is impossible verify discussion. Authors should report it.

The authors apologize for this mistake and have corrected Figure 1 to accurately reflect the figures we submitted separately from the embedded text.

Discussion

Line 382. Authors should explain what they mean as “genetic composition”. From a lexical point of view, the terminology appears quite strange.

We thank the reviewer for pointing this out. We have reworded that sentence to the following in Line 407-408:

“Nacional beans grow in Ecuador and are very similar to Criollo beans.”

References paragraph

The authors should verify the references n. 14, then, it is correct reported in this manuscript they should carefully check. Di Mattia, C,; Sacchetti, G.; Seghetti, L.; Piva, A.; Mastrocola, D. “Vino Cotto” Composition and Antioxidant Activity as Affected by Non Enzymatic Browning. 2007, 19, 413-424. The name of the journal is wrongly reported.

We appreciate the reviewer pointing this out. Reference 14 has been removed from the reference list.